# Current Evidence on Raw Meat Diets in Pets: A Natural Symbol, but a Nutritional Controversy

**DOI:** 10.3390/ani15030293

**Published:** 2025-01-21

**Authors:** Yang Lyu, Caimei Wu, Lian Li, Junning Pu

**Affiliations:** 1Key Laboratory of Animal Disease-Resistance Nutrition, Sichuan Province, Ministry of Agriculture and Rural Affairs, Ministry of Education, Animal Nutrition Institute, Sichuan Agricultural University, Chengdu 611130, China; yang.lyu@sicau.edu.cn (Y.L.);; 2College of Animal Science and Technology, Nanjing Agricultural University, Nanjing 210095, China

**Keywords:** RMBDs, raw meat, gut health, nutrition, dogs, cats

## Abstract

Raw meat diets for pets remain popular due to owners’ pursuit of “natural” choices. Owners report health improvements, but these lack scientific support. Nutritional experts and public health bodies have concerns regarding contamination, imbalances, and sustainability. This review examines the potential benefits and risks of raw meat diets for cats and dogs.

## 1. Introduction

Feeding pets with raw meat products has become a popular topic among pet owners worldwide over the last decade [1,2]. A large survey in the USA from 2017 to 2018 found that 4% of cat owners and 3% of dog owners purchased raw pet food [3]. Morelli et al. conducted a survey of Italian dog owners who fed their dogs raw meat-based diets (RMBDs); 94% of participants believed RMBDs were completely safe, and 80% had entirely eliminated commercial dry and wet pet food [4]. It has been estimated that raw pet food sales have increased by at least 15% annually in recent years [5] and that approximately 10% of cats and 15–25% of dogs are currently fed raw meat [6]. An observational study of pet feeding practices from 2008 to 2018 showed that the proportion of raw feeding rose from 9.6% to 53% [7], and other surveys reported proportions ranging from 37% to 57% [8,9,10]. Despite the majority of public health and veterinary organisations advising against raw meat diets due to the risk of nutritional imbalances and microbial contamination [11,12,13], they remain popular.

Raw meat-based diets (RMBDs) are diets consisting of uncooked ingredients derived from domesticated or wild animals and fed to dogs and cats [2]. RMBDs are often selected for their perceived natural characteristics and health benefits, such as resembling the canine/feline ancestral diets, being unprocessed, composed of whole ingredients, preservative-free, highly digestible, and claimed to improve stool and coat/skin quality [9,14,15,16,17]. However, specialists emphasise that RMBDs also pose potential health risks. For example, homemade and some commercial RMBDs (which are truly raw) have been shown to increase pathogen exposure, can be inconvenient to store and feed, may cause nutritional imbalances without professional formulation and monitoring, and can affect food sustainability [1,2,14,18,19]. Differing attitudes towards RMBDs have generated debate, highlighting the need for up-to-date knowledge and a more in-depth understanding.

While several reviews have discussed the potential benefits and risks of RMBDs, most have focused on nutrient content and bacterial pathogens [20,21,22]. To the authors’ knowledge, only one recent review by Butowski et al. (2022) has outlined the current evidence on the influence of raw meat feeding on the faecal microbiota in dogs and cats [23], and another review on the effect of nutrition on the microbiome briefly mentioned RMBDs [24]. However, these two reviews contain only limited citations focusing on the effects of RMBDs on metabolism, gastrointestinal health, or other health outcomes. Moreover, most RMBD studies lack detailed dietary information or physiological data (e.g., faecal scores, metabolites) necessary to draw robust conclusions about health consequences [23,24]. To provide owners, veterinarians, and researchers with a more comprehensive understanding of the advantages and disadvantages of RMBDs, this review summarises the latest scientific evidence and discusses the potential health benefits and risks associated with these diets.

## 2. Materials and Methods

A comprehensive literature search was conducted using Web of Science (https://www.webofscience.com/ (accessed on 30 August 2024)), ScienceDirect (https://www.sciencedirect.com/ (accessed on 1 January 2023)), Embase (https: //www.embase.com/ (accessed on 30 August 2024)), Scopus (https://www.scopus.com/ (accessed on 30 August 2024)), PubMed (https://pubmed.ncbi.nlm.nih.gov/ (accessed on 1 January 2023)), and Google Scholar (https://scholar.google.com/ (accessed on 1 January 2023)) databases. The search was performed using different combinations of the following keywords: RMBD/raw meat diet, BARF/bones and raw food, dogs/cats/pets, and *Canis familiaris*/*Felis catus*, with a time frame from January 2000 to December 2024. Only articles with titles and abstracts written in English are included. Studies or cases including hypotheses about RMBDs without experiment evidence were excluded, as well as letters, opinions, and mini-reviews. An overview of findings is provided below, and the classification of research evidence was performed using five hierarchy levels [25], as explained in Table 1.

## 3. Definition of Raw Meat Diets

Raw meat diets include uncooked food ingredients, including meat, organs, and bones from mammals, poultry, or fish (farmed or wild animals), uncooked eggs and unpasteurised milk, and some also contain raw vegetables, fruits, and a wide variety of supplements [14,20]. On a dry matter basis, RMBDs typically contain higher levels of protein (>60%) and/or fat (>20%), differing substantially from commercial dry (generally 25–35% protein and 10–25% fat) and wet (generally 35–55% protein) adult maintenance diets [26].

There are two main categories of RMBDs: commercial and home-prepared. Typical commercial types of RMBDs are frozen, fresh, and freeze-dried [27]. These diets are commonly mass-produced in manufacturing facilities or industrial kitchens and are formulated to fulfil specific nutritional needs for all life stages, growth, or adult maintenance [14,22]. However, some of these products might be designed for occasional or complementary feeding purposes and therefore lack complete nutritional value or balance [22]. Another less frequently encountered type of RMBD involves the combination of a premix, comprising grains, fruits, vegetables, vitamins, and minerals, with a raw meat protein source, either as a mixed blend or as a complete, integrated product. This type of diet is intended to provide a complete meal for pets but is only partly raw since processed ingredients are also used [28]. Home-prepared RMBDs are made by owners, often according to various well-known published resources, such as *Bones and Raw Food* (BARF) [29], the *Volhard Diet* [30], the *Ultimate Diet* [31], or other recipes, diets and programmes created by vets, trainers, breeders, and pet owners [32]. However, these recommendations are usually formulated without scientific peer review [33].

## 4. Latest Scientific Data on the Health Effects of RMBDs

Overviews of the available research on the effect of RMBDs on body condition, blood characteristics, the gut microbiome, the gut metabolome, and host metabolism are provided in Table 2 and Table 3 and discussed in detail below.

### 4.1. Body Condition

Consuming a conventional high-protein, low-fat diet with restricted energy has been found to facilitate weight loss and the maintenance of lean body mass in dogs and cats, compared to a lower-protein diet with similar energy intake [34]. Similar characteristics could potentially also be applicable to some high-protein RMBDs, and five Level I studies have reported a stable body condition score and body weight in cats and dogs consuming RMBDs [35,36,37,38,39], which demonstrates the plausible role of RMBDs in maintaining body weight and/or their potential to prevent weight gain.

However, whether the consumption of RMBDs could aid weight loss is sometimes controversial. It is worth noting that many RMBDs have relatively higher fat and lower fibre content, which presents a challenge to typical weight loss diets (e.g., high protein and fibre, low fat) due to their high energy density [34]. On the other hand, reducing dietary carbohydrates by replacing them with high-fat and/or high-protein (e.g., ketogenic or very-low-carbohydrate diets) can increase the dietary energy required to maintain body weight and increase post-absorptive circulating energy availability [40], which may still facilitate weight loss. Body weight and body condition are influenced by food ingredients and their digestibility, which varies across studies. Proponents claim that raw diets commonly possess greater digestibility compared to commercial dry diets; this would lower food intake while maintaining energy needs [38,41]. Nevertheless, digestibility depends on the ingredients used. In several studies claiming that RMBDs have better digestibility, there are significant differences in the ingredients used and nutrient content in the different diets [38,41,42]; these studies are therefore limited. Moreover, existing evidence has shown inconsistent results regarding digestibility, as a few studies have observed higher protein digestibility in high-temperature and high-pressure processed meat compared to the same type of raw meat (details will be discussed later) [42,43]. In addition, the more ideal body condition of cats fed raw versus dry diets may be confounded by the fact that cats eating dry diets are often fed ad libitum, whereas this is not common when feeding RMBDs [44,45]. These need to be further explored in future research.
animals-15-00293-t002_Table 2Table 2Latest scientific data on the health effects of RMBDs in dogs.ReferenceDietsNutrient ContentItemsMajor FindingsLevel[35] Sandri et al., 2016Raw beef with premix vs. dry dietsCP: 26.2 vs. 26.7%Fat: 18.2 vs. 10.6%CF: 9.5 vs. 2.2%BodyMaintained body weight and condition scoreIFaecalLower lactic acid levelMicrobiomeDominated by *Fusobacterium* and *Clostridium*[36] Sandri et al., 2019Raw beef with grain /bean powders vs. dry dietsCP: 27/26 vs. 24%Fat: 19/19 vs. 15%CF: 0.7/0.8 vs. 2.2%BodyMaintained body weight and body condition scoreIFaecalHigher isovalerate levelMetabolitesIncreased cholesterol, myo-inositol, gluconic acid, isomaltose, 4-hydroxybutyric acid, 4-aminobutyric acid and threonic acid[37] Schmidt et al., 2018Commercial BARF vs. dry/wet dietsUnavailableBodyMaintained body weight and condition scoreIFaecalIncreased faecal cholesterol concentrationMicrobiomeHigher dysbiosis index, *C. perfringens*, *Streptococcus*, and *E. coli*[39] Hiney et al., 2021RMBDs vs. dryUnavailableBodyMaintained body weight and body condition scoreI[41] Algya et al., 2018Commercial RMBDs vs. dry/mildly cooked dietsCP: 25 vs. 24/31%Fat: 34 vs. 13/28%CF: 6.9 vs. 9.6/12%BloodHigher chloride, lower TG and ALPIFaecalHigher ammonia, lower pHMicrobiomeLower *Bifidobacterium*, *Turicibacter*, and higher *Fusobacterium*[46] Anderson et al., 2018Raw beef vs. dry dietsCP: 76 vs. 29.9%Fat: 17.9 vs. 27.1%CF: 0.6 vs. 2.4%BloodDecreased cytokine gene and receptor expressionI[47] Anturaniemi et al., 2020Commercial BARFs vs. dry dietsCP: 38 vs. 27.5%Fat: 50 vs. 17.4%CF: 1.5 vs. 1.4%BloodAnti-inflammatory/oxidative gene enhancement of IGHM, IGLL5, CD79BI[48] Kim et al., 2017Various raw meat and vegetables vs. dry dietsUnavailableMicrobiomeMore diverse and abundant microbial compositionI[49] Bermingham et al., 2017Raw beef with mineral premix vs. dry dietsCP: 76.3 vs. 29.0%Fat: 17.9 vs. 27.1%CF: 0.6 vs. 2.4%MicrobiomeDominated by *Peptostreptococcus*, *Fusobacterium*, *Blautia*, *Clostridium* and *Lactobacillus*I[50] Scarsella et al., 2020Commercial BARF vs. dry/homemadeUnavailableMicrobiomeHigher *Clostridiaceae*, *Coriobacteriaceae* and *Fusobacteriaceae*I[51] Beloshapka et al., 2013Raw beef vs. chicken with fibresCP: 25 vs. 32%Fat: 64 vs. 51%CF: 2.6 vs. 4.2%MicrobiomeDominated by *Fusobacterium* and *Clostridium*I[43] Cai et al., 2022Raw vs. pasteurised/high-temperature sterilised dietsCP: 31 vs. 30/30%Fat: 9 vs. 9/9%CF: 1 vs. 0.98/1.0%BloodHigher neutrophils and lower triglycerideIIFaecalLower acetic acid, propionic acid, butyric acidMicrobiomeHigh *Ruminococcaceae* low *Shigella Prevotellaceae*[52] Castañeda et a., 2023Homemade BARF vs. dry dietsUnavailableMicrobiomeHigher *Fusobacterium*, *Bacteroides*, and *C. perfringens*.IINutrient content is on a dry matter basis. CP, crude protein; CF, crude fibre; BARF, bones and raw food; TG, triglyceride; ALP, alkaline phosphatase; IGHM, immunoglobulin heavy constant mu; IGLL5, immunoglobulin lambda-like polypeptide 5; CD79B, immunoglobulin-associated cluster of differentiation 78 beta.
animals-15-00293-t003_Table 3Table 3Latest scientific data on the health effects of RMBDs in cats.ReferenceDietsNutrient ContentItemsMajor FindingsLevel[17] Kerr et al., 2011RMBDs based on beef/bison/elk/horseCP: 66/49/79/60%Fat: 19/38/5.4/26%CF: 7/6.7/9.2/7.1%MetabolismSimilar N retention and maintained N metabolismI[38] Butowski et al., 2019Commercial beef-based RMBDs vs. dry dietsCP: 74/66 vs. 42%Fat: 19/15 vs. 16%CF: 3.5/0.9 vs. 1.8%BodyMaintained body weight and body conditionIFaecalLower propionateMicrobiomeDominated by *Fusobacterium*, *Prevotellaceae Clostridium*, and *Clostridiales*[53] Kerr et al., 2012Commercial beef-based RMBDs vs. dry or beef-based cooked dietsCP: 52.5 vs. 52/57%Fat: 20 vs. 18/17%CF: 4.2 vs. 4.9/4.2%BloodHigher levels of serum albumin and cholesterolI[54] Hamper et al., 2017Commercial RMBDs vs. wet dietsCP: 37 vs. 36%Fat: 26 vs. 29%BloodHigher globulin levelsI[55] Kerr et al., 2014Raw whole chicks vs. dry dietsCP: 71.4 vs. 38.9%Fat: 20 vs. 14.4%MicrobiomeHigher *Fusobacterium* and *Clostridium*I[56] Momoi et al., 2001Commercial fish-based RMBDsUnavailableBloodIncreased plasma lipid peroxideIVNutrient content is on dry matter basis. CP, crude protein; CF, crude fibre.


### 4.2. Blood Characteristics

Some indeterminate effects of RMBDs on blood characteristics have been reported (five Level I, one Level II, and one Level IV evidence). Some evidence indicates that serum biochemical values in dogs and cats fed RMBDs may deviate from laboratory reference ranges. Potential anti-inflammatory effects of RMBDs in dogs were also observed in a few recent gene expression studies. Specifically, compared to dogs fed extruded dry diets, dogs fed a raw beef diet demonstrated a decrease in the expression of pro-inflammatory cytokine genes (i.e., IL2, INFγ, CCL5, and IL15) and receptors (i.e., FOXO1, CD40, and TLR4) [46]. Dogs fed a commercial complete RMBD showed a transcriptomic enhancement of innate immunity (i.e., IGHM, IGLL5, CD79B), which could decrease oxidative stress and inflammation [47], and dogs fed different types of RMBDs had a significantly lower concentration of glycoprotein acetyls, a composite inflammatory marker [57]. Nonetheless, the exact underlying causes of these changes remain unclear, as the composition of the diets studied differed substantially. In cats, RMBDs exceeded the maximum reference levels of blood albumin, globulin, lipid peroxide, and cholesterol [53,54,56], which may be caused by increased dietary protein and fat intake.

In addition, although within reference ranges, some common findings of blood characteristics were observed in multiple studies of RMBDs in dogs. Compared to dogs fed extruded dry diets, two studies documented significantly lower serum alkaline phosphatase [39,41,43] and four studies documented lower blood lipids (i.e., triglycerides, cholesterol) [41,43,56,57] in dogs fed RMBDs. These results imply that RMBDs may promote lipid metabolism compared with extruded dry diets, which might be a potential mechanism by which RMBDs could aid weight loss [39,41,43,57]. However, further research is needed to better understand the underlying mechanisms.

### 4.3. Gut Microbiome

Numerous studies have made great progress in unravelling the exact composition and functionality of the gastrointestinal microbial community in dogs and cats [48,58], and investigations into RMBD feeding are also beginning to emerge. Regarding gut microbial diversity, however, there are apparent inconsistencies in the currently available canine studies, and data are scarce for felines. In dogs fed raw meat compared to those fed dry or wet diets, microbial diversity has been reported to be not different [37], increased [35,48], or decreased [49,50]. These discrepancies may be attributed to the relatively small sample sizes in those studies, different alpha diversity indices used (i.e., Shannon, Chao1, or both), and, particularly, to differences in the dietary composition of the RMBDs. The latter was well illustrated by an observed greater diversity when RMBDs were fed with vegetable ingredients and/or complementary flours and fibre [35,48], as increased plant fibre and carbohydrate may contribute to higher microbial diversity by increasing fermentation activity [23,58].

Distinct differences in microbiome composition have been observed in dogs and cats consuming RMBDs compared with commercial dry or wet food [37,38,41,43,49,50,51,52,55]. In general, dogs fed raw diets have been found to have decreased levels of Bacteroidetes and Firmicutes [49]. Most species in these two phyla are involved in carbohydrate fermentation, and the observed reduction suggests that a dietary transition to low-carbohydrate RMBDs causes compositional and functional changes in the microbiome [59]. In contrast, the abundance of microbial populations associated with protein metabolism was upregulated in dogs who consumed RMBDs long-term, particularly Fusobacteria, Proteobacteria, and the genera *Blautia*, *Clostridium perfringens*, and *Fusobacterium varium*, which produce butyrate from amino acids [43,52,59,60], indicating changes and adaptation to the diet [35,37,48,49]. Notably, higher abundances of *Fusobacterium* and *Clostridium* are often considered detrimental to dogs, as most of their members are widely known pathogenic bacteria (e.g., *Fusobacterium nucleatum*, *C. difficile*, and *C. perfringens*) [61]. This suggests dysbiosis in dogs consuming RMBDs, although scientific evidence is lacking.

Similar results were observed in cats fed RMBDs; the microbiome was dominated by *Clostridium* and unclassified members of Peptostreptococcaceae, *Fusobacterium*, Prevotellaceae, and Clostridiales [38]. Also, greater relative abundances of *Fusobacterium* and *Clostridium* were displayed when fed raw whole chicks compared with an extruded chicken diet [52].

### 4.4. Gut Metabolome

Observed shifts in the faecal metabolome often point towards potential adverse health effects associated with RMBDs. In a study in dogs by Sandri et al., the raw meat diet resulted in lower faecal lactic acid levels [35] and higher isovalerate levels compared to the commercial dry diet [36]. Furthermore, a BARF diet increased faecal cholesterol concentrations compared to the commercial dry diet, but no changes were observed for blood cholesterol, faecal bile acids, or other related physiological parameters. Consequently, researchers were uncertain whether the elevated cholesterol was due to excess dietary fat or microbial synthesis [37]. In cats, one study observed lower propionate on the raw diet, higher succinate on the extruded dry diet, and higher lactate on the raw diet supplemented with fibre [38].

These results demonstrate that RMBDs caused shifts in metabolic functions, influencing colonic fermentation of protein and amino acids. However, this cannot be fully confirmed, as the macronutrient content and ingredient composition of the diets in these studies differed substantially. Nevertheless, it can be hypothesised that the observed changes are due to the relatively higher amounts of protein and fat in RMBDs and may also be due to the ingredients used (vegetables or fruit, organs versus muscle, supplements, etc.) and the differences in processing between the various studies. It should, however, be noted that most of these interpretations were extrapolated from human studies, and that the actual health effects of these shifts may vary in dogs and cats since they are better adapted to consuming more protein and fat to meet their unique nutritional requirements compared with humans [58]. Moreover, the aforementioned canine diets were not raw in the strictest sense since processed ingredients, particularly starch sources, were included, which may affect the gut metabolome [35,36,37]. Therefore, conclusions drawn from those studies should not be extrapolated to fully raw diets.

### 4.5. Host Metabolism

Very few studies have investigated the metabolic changes following the consumption of RMBDs in dogs and cats, and consequently, no sound conclusions can be drawn at this time. Only one study investigated differences in nitrogen metabolism in cats fed RMBDs, in which nitrogen retention was similar to that of cats fed traditional extruded dry diets [17]. Despite differences in protein concentrations and nitrogen intake, cats maintained their nitrogen balance [17].

## 5. Owner Motivations for Feeding Raw Meat Diets

Despite the lack of definitive scientific evidence and professional recommendations, a growing number of pet owners are choosing to transition away from commercially available dry/wet diets, often recommended by veterinarians. This shift reflects evolving attitudes towards pet food and feeding practices [4,19]. Seeking “natural” diets, pet owners often believe in the anecdotal health benefits of raw, whole-food diets, especially the absence of processing and the inclusion of whole foods rather than food fractions [15,16,17]. The belief that “animals cannot cook” or that “there is no kitchen in nature” fuels the perception that RMBDs are better suited to the purported “wild carnivore nature” of their pets, as an RMBD is the diet that most closely resembles the canine ancestral diet on which canines have evolved over millions of years [14,15].

Cost is another factor in the choice of home-prepared raw diets, often perceived as lower than commercial raw foods. Importantly, many pet owners approach feeding their pets similarly to how they feed their own families, encompassing diverse social and cultural aspects [4,10]. This is evident in varied food preferences, for example, the non-uniform dietary habits in China [62], halal diets in Muslim countries [63], and the generally meat-rich diets prevalent in Western countries, all of which can influence motivations for feeding RMBDs.

Home-prepared RMBDs frequently involve rotating multiple animal ingredients and by-products (e.g., organs and bones) [21]. This rotation is believed to provide a diverse range of nutrients, including essential fatty acids, amino acids, minerals, and vitamins [29,30,31,32], potentially resulting in a complete and balanced diet and beneficial health effects. This belief in rotational variety is not unique to RMBD proponents; some owners who feed complete and balanced commercial foods also endorse this concept [7,8,9,10,14]. Furthermore, some owners express increased concern and distrust regarding traditional extruded dry pet food, particularly due to perceived ambiguities in ingredient lists [4] and the occurrence of several pet food recalls in recent years [64,65].

## 6. Benefits of Raw Meat Diets—Public Claims

From the pet owner’s perspective, the popularity of RMBDs and their perceived safety and health benefits stem largely from various public claims [4]. Proponents suggest that RMBDs may enhance digestibility, improve coat and stool quality, aid in teeth cleaning, and reduce the risk of nutritional imbalances [21,66]. While some research evidence supports these claims, the level of evidence is generally low (two Level I, one Level II, six Level III, and three Level IV studies, Table 4).

Although ingredient details were not fully provided, a recent canine study found an increased food digestibility coefficient (95.7 ± 1.2% vs. 57.1 ± 9.1%) 45 days after switching from an extruded dry diet to a raw meat diet with comparable ingredients [42]. Another study showed significantly higher fat digestibility (97.5% vs. 92.1%) with a raw diet (raw chicken and vegetables mixed with premixes) compared to a commercial dry diet [41]. A raw beef diet (beef, bones, and premixes) had a significantly higher apparent digestibility of dry matter (93.8 vs. 79.6%), gross energy (98.4 vs. 80.5%), protein (99.3 vs. 79.5%), and fat (99.6 vs. 91.0%) [38]. Improved stool quality has also been observed in boxer dogs fed a raw beef diet with a supplement [35] and in kittens fed a rabbit-based raw diet [67], when compared to extruded dry diets. Cleaner teeth were reported by 94% of owners in a survey who fed RMBDs to their pets [4]. Moreover, dogs fed RMBDs showed improved coat scores when compared to dogs fed extruded dry diets [39], and cats fed a diet consisting of whole rabbits also appeared to have improved coat quality [67]. A few studies suggest that dogs fed RMBDs may have a lower risk of developing certain diseases and exhibit more beneficial metabolic effects compared to dogs fed extruded dry diets. This includes fewer cases of atopic dermatitis, chronic enteropathy, and allergy-related skin signs as puppies mature [68,69,70,71,72], reduced risk of calcium oxalate uroliths in adult dogs [73], and more favourable metabolic profiles in dogs with atopic dermatitis (e.g., lower serum sulphur-containing compounds, such as methionine and cystathionine) [74].
animals-15-00293-t004_Table 4Table 4Evidence regarding possible or perceived benefits of RMBDs.ReferenceDietsClaimed Benefit/AspectLevel[36] Sandri et al., 2019RMBDs vs. dry diets in dogsBetter stool qualityI[38] Butowski et al., 2019Commercial RMBDs vs. dry diets in catsHigher apparent digestibility of dry matter, gross energy, protein, fatI[74] Moore et al., 2020Commercial RMBDs vs. dry diets in dogsBeneficial for metabolic healthII[57] Puurunen et al., 2022Raw vs. other diet types in dogsBeneficial for lipid metabolism, lower inflammatory status.III[68] Hemida et al., 2021Non-processed meat-based vs. ultra-processed carbohydrate-based dietsLower IBD risk in puppiesIII[69] Hemida et al., 2021Raw meat/organs vs. dry in Finland dogsLower risk of the development of allergy/atopy skin signsIII[70] Hemida et al., 2020Non-processed meat-based vs. ultra-processed carbohydrate-based dietsLower risk of the development of atopic dermatitisIII[71] Vuori et al., 2023Non-processed meat-based vs. ultra-processed carbohydrate-based dietsLower risk of the development of chronic enteropathyIII[72] Nijsse et al., 2016Frozen raw meat vs. commercial dry dietin dogs in NetherlandsLower risk of *T. canis* infectionsIII[4] Morelli et al., 2019Multiple RMBDs (no control)Cleaner teethIV[67] Glasgowet al., 2018Whole-rabbit in kittens (no control)Better stool qualityBetter coat qualityIV[73] Dijcker et al., 2012Multiple RMBD vs. dry diets in dogsLower calcium oxalate urolith riskIV


However, the majority of the aforementioned findings remain largely anecdotal, opinion-based, or hypothetical, lacking robust scientific support. This lack of scientific backing is reflected in critical reviews and guidelines from professional veterinary organisations and specialists [7,8,9,10,11,12,13,75]. In fact, many purported benefits, such as improved digestibility, stool quality, and reduced disease risk, are often ambiguous, as the actual cause may lie in the specific dietary ingredients selected rather than the raw nature of the diet itself. Similarly, the claim of reduced nutritional imbalance is easily influenced by the quality of ingredients and social/cultural factors, making it difficult to isolate the true effect of the raw diet from other dietary elements. Crucially, significant differences in the nutritional composition between RMBDs and control diets (e.g., dry, wet) often exist in the studies, blurring the line between the effect of different nutrient content and the effect of the raw diet type itself. While formulating raw diets with similar nutritional content to traditional diets is challenging, this critical comparison must be addressed in future academic research. Such studies should meticulously consider the specific nutrient profiles and ingredients employed to accurately assess the essential differences between diet types.

## 7. Risks of Raw Meat Diets

From the point of view of veterinarians, generalists, and specialists, concerns with RMBDs include pathogenic hazards, nutritional imbalance, and sustainability, which are further discussed below.

### 7.1. Pathogenic Hazards

Pathogenic hazard remains a significant concern in pet food, highlighted by specialists and major public health organisations [2,11,12,13]. In this section, 4 Level II, 33 Level III, and 24 Level IV studies are cited to summarise pathogenic contamination issues in RMBDs (Table 5). Various studies have found that between 6% and 20% of raw meat products are contaminated with *Salmonella* [76,77,78,79,80,81,82,83,84,85,86,87,88,89,90,91,92,93]. Considerable levels of Enterobacteriaceae contamination have been found in commercial RMBDs (3.61 × 10^6^–8.39 × 10^6^ CFU/g), and a higher count of Enterobacteriaceae was detected in frozen poultry RMBDs compared to partially thawed products [19,94]. Moreover, approximately 60% of commercial RMBDs have been found to be contaminated with *Escherichia coli* [95,96,97,98,99,100,101,102,103]. *Campylobacter* contamination has been noted in RMBDs as well, particularly based on poultry meat and by-products [78,81,82,104,105,106]. Raw meat feeding has also been associated with infection of *Mycobacterium bovis* in dogs and cats, leading to tuberculosis in a few cases [107,108,109]. In addition, when consuming raw meat from poultry or wild birds, dogs and cats are highly likely to contract avian influenza (i.e., H5N1, H5N6), which is currently an ongoing outbreak with several reported cases and multiple deaths [110,111,112,113]. Other microbial pathogens in RMBDs included *C. perfringens* [114,115], *C. difficile* [114,115,116,117], *Staphylococcus aureus* [83,104], *Listeria* [80], and *Brucellosis* [118,119], which are relatively minorly detected and can be reduced through commonly used sterilisation methods. Protozoal contamination is also a concern; feeding cats raw meat has been linked to infections caused by *Toxoplasma gondii*, and *Cryptosporidium* spp. was detected in two commercial RMBDs [120,121,122,123,124].

Many pet food packages lack warnings about the proper preparation and hygienic handling of the food, while some packaging may also be defective and prone to leaks [125,126]. Furthermore, the risk of infectious disease associated with RMBDs is not limited to the pets themselves [127]. Pets fed raw food diets can become carriers of pathogens, such as *Salmonella* spp. and *T. gondii*, a risk that is often underestimated by RMBD proponents [128,129,130]. Consequently, owners are also at risk due to frequent contact with pathogen-carrying pets and contaminated pet food (i.e., purchase, storage, preparation, and feeding) [14]. Such risks are greatest for immunocompromised humans, such as those who are very young, old, with comorbidities or on immunosuppressive (e.g., systemic steroidal) therapy [131], but not eliminated for others, e.g., who may come into contact during dog walks [131,132]. Moreover, livestock-derived bacteria, including both pathogenic and commensal species, can carry antibiotic resistance genes, including resistance to critically important antibiotics like extended-spectrum cephalosporins [94,133,134,135,136]. Feeding raw diets to pets, therefore, increases the risk of these animals shedding resistant bacteria, thereby contributing to the spread of antibiotic resistance [2]. Notably, wild meat presents a lower risk for antimicrobial resistance issues than livestock [137], and evidence regarding microbial contamination seems to be presented more often in white meat (e.g., fish, chicken…) than in red meat (e.g., beef, pork…) [77]. The meat species of origin should therefore also be a consideration when owners choose products.

Overall, microbial contamination is the most significant challenge associated with RMBDs. For loyal advocates of such diets, it is highly recommended that preventative hygienic measures be taken during both RMBD preparation and feeding. It is also advisable to choose products that have undergone sterilisation techniques such as high hydrostatic pressure, which is the most commonly used and well-proven method for pathogen mitigation in raw pet foods [138]. High hydrostatic pressure in the range of 300–600 MPa for a short period can, for example, inactivate up to 5-log units of vegetative pathogenic and spoilage microorganisms, including *E. coli*, *Campylobacter*, *Pseudomonas*, *Salmonella*, *Listeria*, and *Yersinia* [138,139,140]. Other potential methods to reduce the proliferation and transmission of foodborne pathogens include the use of organic acids (e.g., citric, acetic, or lactic acid), bacteriophages, ozone, and protective (probiotic) cultures [141,142]. Beyond the recognised risk of microbial contamination, other pathogenic hazards associated with the feeding of raw meat to pets include parasites (e.g., intestinal worms) and prion proteins (i.e., implicated in bovine spongiform encephalopathy and variant Creutzfeldt–Jakob disease in humans) [126,143,144]. Despite being less frequently documented and/or studied, these potential risks deserve attention.
animals-15-00293-t005_Table 5Table 5Evidence of RMBDs regarding pathogenic hazard risks.ReferenceRisksDietsLevel[84] Joffe et al., 2002*Salmonella* in diets and dogsA homemade BARF vs. dry dietII[94] Baede et al., 2017*Enterobacteriaceae* in pet foodsCommercial RMBDs vs. various dietsII[95] Strohmeyer et al., 2006*E. coli*, *Salmonella*, *Cryptosporidium* in pet foodsCommercial RMBDs vs. dry/wet dietsII[101] Groat et al., 2022*Salmonella* and antimicrobial-resistant *E. coli* in UK dogsMultiple RMBDs vs. various dietsII[27] Mehlenbacher et al., 2012*Salmonella* in raw pet food in St. PaulMultiple commercial RMBDsIII[77] Finley et al., 2008*Salmonella* and antimicrobial resistance in Canadian canine foodMultiple commercial RMBDsIII[78] Hellgren et al., 2019*Salmonella*, *Campylobacter*, *Clostridium*, *Enterobacteriaceae* in dogsMultiple RMBDsIII[79] Solís et al., 2022*Salmonella*, *Listeria*, and *Campylobacter* in canine diets and faecesCommercial RMBDs vs. dry dietsIII[80] Nemser et al., 2014*Listeria* and *Salmonella* in pet foodsCommercial RMBDs vs. various dietsIII[81] Lenz et al., 2009*Campylobacter* and *Salmonella* in pet dogsMultiple commercial RMBDsIII[82] Cammack et al., 2021*Salmonella*, *Campylobacter*, and *E. coli* in pet foodsMultiple commercial RMBDsIII[83] Azza et al., 2014*E. coli*, *Salmonella*, *Staph aureus* in pet foodsMultiple RMBDsIII[88] Finley et al., 2007*Salmonella* shedding in research dogsCommercial RMBDsIII[89] Leonard et al., 2011*Salmonella* in pet dogs in CanadaMultiple RMBDs vs. various dietsIII[90] Lefebvre et al., 2011*Salmonella* and *E. coli* in therapy dogs in CanadaMultiple RMBDs vs. various dietsIII[91] Morley et al., 2006*Salmonella* infections at Greyhound breeding facility in US.Multiple RMBDsIII[92] Pitout et al., 2003Antimicrobial-resistant *Salmonella* in Canadian pet treatsPet treats containing dead raw beefIII[96] Gibson et al., 2022*E. coli* in pet foodsCommercial RMBDs vs. dry/wet dietsIII[97] Nilsson et al., 2015Antibacterial-resistant *E. coli* in Swedish canine foodsCommercial RMBDs included poultryIII[98] Treier et al., 2021Shiga toxin-producing *E. coli* in raw pet foodsMultiple commercial RMBDsIII[102] Naziri et al., 2016*E. coli* in dogs and humans in IranMultiple RMBDs vs. various dietsIII[103] Wedley et al., 2017Antimicrobial-resistant *E. coli* in UK dogsMultiple RMBDs vs. various dietsIII[106] Parsons et al., 2011*Campylobacter* infection in kennelled dogs in UKMultiple RMBDsIII[107] O’Halloran et al., 2019Tuberculosis due to *M. bovis* in six pet cats in UKMultiple RMBDsIII[108] Phipps et al., 2018*M. bovis* outbreak in foxhound, working dogs, and humans in UKUnspecifiedIII[109] Roberts et al., 2014*M. bovis* infection in cats in UKUnspecifiedIII[110] Lee et al., 2018Avian Influenza A(H5N6) infection in cats in South KoreaUnspecifiedIII[111] Marschall et al., 2008Avian Influenza H5N1 infection in cats in Germany and AustriaUnspecifiedIII[114] Weese et al., 2005*C. perfringens* and *C. difficile* in Canadian pet foodsMultiple RMBDsIII[123] Dubey et al., 2005*T. gondii* in meat and cats in USBeef, chicken, and pork in storesIII[124] Jokelainen et al., 2012*T. gondii* in Finish catsRMBDs vs. various dietsIII[125] Bojanić et al., 2017*Campylobacter* in dogs and homemade RMBDs in New ZealandMultiple homemade RMBDsIII[115] Viegas et al., 2020*Salmonella*, *C. perfringens* and *C. difficile* in dogs in BrazilRMBDs vs. various dietsIII[130] Reimschuessel et al., 2017*Salmonella* in US dogs and catsRMBDs vs. various dietsIII[134] Mounsey et al., 2022Antibacterial-resistant *E. coli* in UK dogsRMBDs vs. various dietsIII[135] Leonard et al., 2015Antimicrobial-resistant *Salmonella* and *E coli* in dogsRMBDs vs. various dietsIII[137] Nuesch et al., 2019*E. coli*, *Salmonella* and antibiotic resistance in Switzerland pet foodsMultiple RMBDsIII[19] Vecchiato et al., 2022*Enterobacteriaceae*, *Salmonella* spp. in Germen dogs and catsMultiple RMBDs (no control)IV[76] Wright et al., 2005*S. Typhimurium* in 4 animal facilities in USAUnspecifiedIV[85] Brisdon et al., 2006*Salmonella* outbreak in humans associated with pet treats in USPet treats containing raw meatIV[86] Behravesh et al., 2010Human *Salmonella* infections linked to dog and cat foodsCommercial RMBDs vs. dry dietsIV[87] Canada, 2020*Salmonella* and *Enterobacteriaceae* in dog treats in CanadaDog treats containing raw meatIV[93] Cavallo et al., 2015Human *S. Typhimurium* outbreak in pet treats in USPet treats containing raw chickenIV[99] Kaindama et al., 2021Shiga toxin-producing *E. coli* in homemade raw dog foodsRMBDs include tripe (no control)IV[100] Jones et al., 2019*Salmonella*, *E. coli*, and *Listeria* in pet diets and faecesCommercial RMBDs (no control)IV[104] Suzuki et al., 2009*Campylobacter* in pet food with poultry meat and by-productsCommercial RMBDs (no control)IV[105] Campagnolo et al., 2017*Campylobacter jejuni* infection in a puppy in USCommercial RMBDs (no control)IV[112] Songserm et al., 2006Avian Influenza A(H5N1) infection in a pet cat in ThailandRaw pigeonIV[113] Yu et al., 2015Avian Influenza H5N6 infection in a domestic cat in ChinaWild birdsIV[115] Bouttier et al., 2010*C. difficile* in commercial feline raw diets in FranceGround meat-based RMBDsIV[117] Rodriguez et al., 2007*C. difficile* infection in dogs in CanadaGround meat-based RMBDsIV[118] Frost et al., 2017*Brucella suis* infection in dogs in UKUnspecifiedIV[119] Mor et al., 2016*Brucella suis* infection in dogs in AustraliaRaw feral pig meat (no control)IV[120] Brennan et al., 2020*T. gondii* in cats in AustraliaRaw chicken/kangaroo/beef (no control)IV[122] Dubey et al., 2003*T. gondii*, *Sarcocystis* spp., *H.heydorni*-like parasite in 3 puppiesRaw tissue of sheep (no control)IV[126] van Bree et al., 2018*E. coli*, *L. monocytogenes*, *S. cruzi*, *S. tenella*, *T. gondii* in Dutch petsCommercial RMBDs (no control)IV[127] Clark et al., 2001*Salmonella* in dogs in CanadaRaw pig ears (no control)IV[128] Binagia et al., 2020*Salmonella* mesenteric lymphadenitis in two dogsCommercial RMBDs (no control)IV[129] Fauth et al., 2015*Salmonella* bacteriuria in a catCommercial RMBD (no control)IV[133] Bacci et al., 2019*E. coli*, *Salmonella* and antibiotic resistance in pet foodsFresh poultry, pork, beef (no control)IV


### 7.2. Nutritional Imbalance Risks

Nutritional imbalances associated with RMBDs have garnered the professional attention of veterinarians and researchers, with three Level III and eight Level IV studies being discussed in this section (Table 6). Evaluations of both commercial and home-prepared RMBDs have identified several nutritional issues, including excessive fat and/or protein content, calcium–phosphorus imbalances, and deficiencies in specific minerals (e.g., potassium, zinc, and magnesium) and vitamins (e.g., vitamin D) [19,26,28]. Several nutritional diseases in dogs and cats have been linked to RMBD consumption [145], such as hypervitaminosis A in a cat fed a pork liver-based raw diet [146], abnormal bone mineralisation in a puppy fed a homemade RMBD with cow and goat dairy products [147], nutritional osteodystrophy in two puppies fed a BARF diet [148], and low plasma taurine levels in numerous dogs and cats tested [149]. However, even when the diet was considered balanced, calcium could not be ingested in some cases, as young puppies are unable to crush bones.

Home-prepared diets can inherently lead to nutritional imbalances and deficiencies [32,33]. In practice, owners may be tempted to simplify recipes, and wrongly assume that rotating multiple ingredients can counteract nutritional imbalances [22]. Moreover, recipes for pet meals may include poorly defined ingredients and quantities, making it difficult for owners to follow them accurately. Additionally, some ingredients may not be locally available or may vary in quality, further complicating the preparation process [2]. A common shortcoming of homemade RMBDs is the lack of wet chemistry nutritional analysis to assess and assure nutritional balance. These diets are often constructed without evaluation through feeding trials or by nutrition specialists, relying on estimations rather than analysis [2]. While specialists have observed and actively warned about the nutritional risks of RMBDs, the supporting scientific evidence is generally low (Levels III and IV) and mainly based on individual cases. Pet food manufacturers have also recognised this issue, and nutritionally balanced RMBDs are now available on the market.

The ingredients used and overall product quality can also lead to other health issues. For example, RMBDs that include gullets and thyroids have been linked to increased serum thyroxine concentrations or clinical signs of hyperthyroidism in 15 dogs (three Level IV cases), with thyroxine concentrations returning to normal after a dietary change [150,151,152]. RMBDs containing intact bones (e.g., BARF) can cause fractured teeth and gastrointestinal injury, as bones have the potential to create blockages or punctures in the oesophagus and gastrointestinal tract [14,22]. Four Level IV studies reported that bones accounted for 30% to 80% of oesophageal foreign bodies in cats and dogs (Table 6) [153,154,155,156]. Advocates of feeding raw bones claim that raw bones pose fewer issues compared with cooked bones [29], although research on this has, to our knowledge, not been conducted [22].
animals-15-00293-t006_Table 6Table 6Evidence of RMBDs regarding risks of nutritional imbalance and oesophageal foreign body.ReferenceRisksDietsLevel**Nutritional Imbalance**[19] Vecchiato et al., 2022Higher fat, Ca and/or P content than the legal limitsMultiple commercial RMBDs in GermanyIII[149] Hajek et al., 2022Lower plasma taurine in pet dogsMultiple RMBDs vs. dry dietsIII[147] Dodd et al., 2021Abnormal bone mineralization in a puppyHomemade RMBD with cow and goat dairyproducts (no control).IV[26] Dillitzer et al., 2011Vitamin and mineral deficiencies in adult dogsCommercial BARFs (no control)IV[28] Taylor et al., 2009Diffuse osteopenia and myelopathy in a puppyPremix and raw ground beef (no control)IV[145] Lenox et al., 2015Metabolic bone disease in a kittenRaw chicken diet (no control)IV[146] Polizopoulou et al., 2005Hypervitaminosis A in a catHomemade raw pork liver (no control)IV[148] Delay et al., 2002Nutritional osteodystrophy in puppiesA commercial BARF (no control)IV[150] Zeugswetter et al., 2013Hyperthyroidism in two dogsCommercial RMBDs with thyroids (no control)IV[152] Sontas et al., 2014Hyperthyroidism in a miniature pinscher bitchHomemade BARF included cattle bones andmeat of the head and neck region (no control)IV**Oesophageal Foreign Body**[153] Rousseau et al., 2007Bones lodged in oesophagus in 48 dogs with esophagitisUnspecified BARFIV[154] Gianella et al., 2009In 102 dogs with foreign bodies, 50 are bonesUnspecified BARFIV[155] Frowde et al., 2011Five cats had oesophageal foreign bodies, 2 are bonesUnspecified BARFIV[156] Thompson et al., 201129.7% of oesophageal foreign bodies were bones in dogsUnspecified BARFIV


### 7.3. Sustainability

Food and feed sustainability refers to the provision of adequate and safe nutrition to end users, while also considering the ecological, economic, and social aspects of sustainability [157]. The formulation of commercial pet foods is often influenced by consumer preferences rather than strictly adhering to nutritional needs. Consequently, these pet foods frequently contain excessive nutrients beyond minimum recommendations, utilise ingredients that could otherwise be used for human consumption, and lead to pets consuming these ingredients in excess. This overconsumption contributes to food wastage and environmental concerns [157].

Protein is the most expensive macronutrient, both economically and environmentally [157]. The recommended minimum allowance of crude protein on a dry matter basis for adult cats is 25% (FEDIAF 2024, European Pet Food Industry Federation) [158] to 26% (AAFCO 2014, Association of American Feed Control Officials) [159], and for adult dogs, it is 18% (FEDIAF 2024 and AAFCO 2014) [158,159]. However, RMBDs typically contain a relatively higher amount of meat, with a protein content well above the recommended allowance for both dogs and cats [160]. Consequently, such protein overconsumption leads to avoidable food waste and environmental pollution (e.g., nitrogen emissions). Additionally, compared with plant-based proteins, animal-based proteins may exacerbate environmental impacts. Data on animal proteins reveal an approximately 11 times larger carbon footprint, a 100 times higher water requirement, 6–20 times more fossil fuel requirement, and 6–17 times more land use compared to plant proteins [129,130,131]. Furthermore, fish-based proteins seem to demonstrate a greater environmental burden than livestock; wild-caught fish require more fossil fuels, and aquaculture-based fish require more land and release biocides or nutrients [161,162,163]. A recent study estimated that globally, dogs and cats consume approximately 7.7% and 1.2% of livestock animals, respectively [164]. A complete transition to nutritionally sound vegan diets worldwide could potentially spare 7.0 billion terrestrial livestock animals from slaughter annually, reduce greenhouse gas emissions by 1.3%, and conserve food energy sufficient to feed 160–190% of the existing pet population or 7% of the global human population [164].

Additionally, due to the demand for food hygiene and freshness, RMBDs may incur higher costs for manufacturing, packaging, transportation, and storage. Although this aspect lacks investigation, professionals involved in the pet food industry have a significant opportunity to enhance the sustainability of pet foods through product design, manufacturing optimisation, public awareness, and policy development.

## 8. General Discussion

While arguments for health benefits versus risks remain controversial, the increased prevalence of feeding RMBDs has become an unstoppable trend [24]. The biggest shortcoming in this context is that several publicly claimed health benefits lack a solid scientific basis, as they are usually based on opinions or hypotheses extrapolated from human studies [2]. On the other hand, many experimental data clearly highlight the risks associated with RMBDs. However, most of these studies were originally designed to criticise or scrutinise food safety issues of RMBDs, and few peer-reviewed studies have attempted to investigate solutions or discuss similar issues in other diet types. Despite the emergence of more research evidence on the health effects of RMBDs, the vast majority of these studies compared RMBDs to conventional diets that have markedly different ingredients and nutrient contents. The results and conclusions obtained were likely attributable to the distinct variations in ingredient and nutrient composition, rather than the diet being RMBD or not; these differences in ingredients and macronutrient profiles are exactly the elements that should have been of concern in those studies. If used ingredients and nutrient composition are similar, the most prominent difference between raw and other diets is likely the heat processing and/or related Maillard reactions [165,166], which may determine the differences found in these studies but have not yet been investigated in pet foods. As heat processing is a major factor in the formation of Maillard reaction products in traditional pet food, and RMBDs are obviously not heat processed, future studies should investigate the effect of heat processing on the formation of Maillard reaction products, using diets with similar nutrient composition and ingredients [165,166]. Moreover, as with traditional diets (e.g., dry, wet, semi-moist), there is also high variability within RMBDs (e.g., BARF vs. whole prey, refrigerated vs. frozen/freeze-dried, etc.) [167], but most studies have only investigated a few RMBD types (i.e., refrigerated/frozen, BARF), which skews findings and perceptions of their benefits and risks. In addition, some opponents and critics have emphasised that the potential health risks are based on several clinical case reports [17,20]. Nevertheless, considering individual differences in specificity and susceptibility, and the very small number of animals and cases, these case reports have limited reference value and should not be considered strong evidence when discussing health effects.

This review summarises the most recent research evidence to comprehensively assess the known and unknown aspects of the health effects of feeding raw meat-based diets (RMBDs). Briefly, RMBDs may promote healthy body weight and condition, and improve stool quality [35,36,37,38,39,67]. Furthermore, RMBDs may elicit anti-inflammatory and antioxidant effects, potentially linked to alterations in gene expression (as evidenced in serum), increase the metabolism and colonic fermentation of protein and amino acids [46,47,53,54,56,57], and result in lower bacterial diversity, with a microbiome dominated by *Fusobacterium* and *Clostridium* and associated fermentation end-products [36,37,38,49,50,51,52,55]. Serum biochemical values may deviate from laboratory reference ranges in dogs and cats fed RMBDs, but the implications of these deviations remain unclear [46,47,53,54,56,57]. However, the current research on RMBD feeding is insufficient to provide a comprehensive nutritional understanding. Further research is needed to explore various aspects, including the specific metabolic changes, host–microbial interactions, potential dose–response effects on inflammation and immune responses, maintaining nutritional balance with added carbohydrates and fibre, and developing strategies for reducing microbial contamination and improving food and feed sustainability.

Owners who favour RMBDs often perceive it as a “natural” feeding method, emphasising the unprocessed, whole-ingredient nature and the supposed wild carnivore nature of their dogs and/or cats. However, dogs have evolved significantly from their wild ancestors and have adapted to a higher carbohydrate metabolism to meet their nutritional requirements [22]. Beyond microbial contamination concerns during purchase and storage, numerous specialists and public health organisations have raised nutritional concerns. Owners are advised to follow veterinarian and nutritionist recommendations and to monitor their pets’ health both short-term and long-term when feeding RMBDs. This includes paying particular attention to supplementing carbohydrates and/or dietary fibre, tailored to the pet’s specific physiological needs. To increase knowledge of the health effects of raw meat diets, future research should focus on comparing a wider range of RMBDs to traditional dry/wet diets with similar ingredient and nutrient profiles. Research should also explore more sustainable ingredients and formulations to enhance diet safety and quality. Pet food manufacturers can mitigate microbial contamination and promote nutritional sustainability by establishing higher hygiene standards, exploring novel sterilisation techniques, designing sustainable manufacturing processes, and optimising raw material selection. Veterinarians and nutritionists who support or oppose RMBDs should play an active role in guiding owners considering feeding RMBDs. Increased pet food variety may improve animal health, pet well-being, and the pet food industry. Scientists should provide more robust scientific evidence, and veterinarians/nutritionists should guide RMBD feeding decisions based on their expertise.

## 9. Conclusions

In summary, limited current evidence in dogs and cats has suggested that feeding RMBDs may lead to a healthy body weight and condition, improved stool quality, compositional and functional changes in the gut microbiome, upregulated metabolism of protein and amino acids and/or fat, and may elicit anti-inflammatory and antioxidant potentials. However, RMBDs also carry considerable risks, including pathogenic hazards (e.g., bacterial, protozoal, influenza), nutritional imbalances (e.g., high-fat content, vitamin and mineral deficiencies), the possibility of oesophageal foreign bodies (i.e., bones), and sustainability issues (e.g., excessive meat consumption, environmental pollution). Crucially, the substantial variations in the types, processing methods, storage, ingredients used, nutrient content, animal dietary habits, and individual differences may significantly influence the health effects and risks associated with RMBDs. These aspects are rarely studied in detail and, as such, further investigations are urgently required.

## Figures and Tables

**Table 1 animals-15-00293-t001:** Levels of evidence hierarchy.

**Level** **I**	High-quality randomised trial or prospective study; testing of previously developed diagnostic criteria on consecutive patients; sensible costs and alternatives; values obtained from many studies with multi-way sensitivity analyses; systematic review of Level I randomised controlled trials and Level I studies.
**Level II**	Lesser quality randomised controlled trial; prospective comparative study; retrospective study; untreated controls from a randomised controlled trial; lesser quality prospective study; development of diagnostic criteria on consecutive patients; sensible costs and alternatives; values obtained from limited studies; with multi-way sensitivity analyses; systematic review of Level II studies or Level I studies with inconsistent results.
**Level III**	Case-control study (therapeutic and prognostic studies); retrospective comparative study; study of non-consecutive patients without consistently applied reference “gold” standard; analyses based on limited alternatives and costs and poor estimates; systematic review of Level III studies.
**Level IV**	Case series; case-control study (diagnostic studies); poor reference standard; analyses with no sensitivity analyses.
**Level V**	Expert opinion.

Adapted from Elsevier—https://scientific-publishing.webshop.elsevier.com/research-process/levels -of-evidence-in-research/ (accessed on).

## Data Availability

Not applicable.

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
