# Peer review of "Current Evidence on Raw Meat Diets in Pets: A Natural Symbol, but a Nutritional Controversy"

_animals, 2025, doi:10.3390/ani15030293_

Round 1
Reviewer 1 Report
Comments and Suggestions for Authors
1. Introduction
This includes the Methodology. That should be removed to a separate section ‘Methodology’.
Unfortunately, the methodology is currently very weak. It is not sufficiently strong to support the claim that this article (adequately) reviews and summarises ‘Current evidence on raw meat diets in pets’ based on a ‘comprehensive literature search.' The following methodological deficits need correction, before that could be considered true. The current draft could be a foundation for a future article, but the literature search needs to be repeated, and it needs to comply with good practice as described below.
First, it uses only 1 reliable database (Google Scholar is not reliable). (Readers could also reasonably be concerned about any use of Google in China being more restricted than outside China). This is not adequate. Best/good practice for database selection while conducting systematic reviews is described in publications such as the below and should be adhered to:
https://systematicreviewsjournal.biomedcentral.com/articles/10.1186/s13643-017-0644-y
https://onlinelibrary.wiley.com/doi/full/10.1002/jrsm.1378
Any language restrictions should be specified.
The search terms are not adequate. Normally, a much more comprehensive list of terms is given. E.g. it’s not clear the current search would have detected ‘raw meat feeding of dogs’ and it would have missed Canis familiaris/Canis lupus familiaris/Felis catus.
These deficits are indicated by the list of retrieved studies in Table 4 being significantly shorter than this list https://sustainablepetfood.info/meat-based-diets/studies-raw-meat/
2. Definition of Raw Meat Diets
“differing substantially from commercial dry/wet adult maintenance diets” – it would be helpful to give an indicative normal range for these.
3. Latest Scientific Data on the Health Effects of RMBDs
Table 2. before each citation provide author, year – so readers can quickly see, e.g. ‘Smith, 2020 [xx]’. Pls do the same for Tables 3-4.
Consider two tables: 1 for dogs, 1 for cats
[38] – what is inulin?
5. Benefits of raw meat diets – public claims
“only 2 level-I, 1 level-I” – the latter should be level-II.
Table 3. Evidence regarding the benefits of RMBDs.
Needs to be renamed to
Table 3. Possible benefits of RMBDs.
Because several are not good quality evidence (as you note in the following text).
Re:
“the definition of the ‘balanced diet’ is easily influenced by the quality of ingredients and social and cultural aspects”
This needs to be reworded, because the definition of a nutritionally a complete and balanced diet is one that meets the nutritional requirements of dogs/cats, and those requirements are published by authorities such as FEDIAF, AAFCO, NRC. These biological requirements are independent of social and cultural aspects, etc.
6. Risks of raw meat diets
6.1 Microbial pathogens
The list of studies showing microbial contamination is long, yet significantly incomplete. More than 80 studies have demonstrated pathogenic hazards assoc with raw meat diets (https://sustainablepetfood.info/meat-based-diets/studies-raw-meat/).
i. This should at least be mentioned, even if not all are cited.
ii. pathogenic hazards are not limited to microorganisms (such as bacteria or protozoa) and can include influenza (avian influenza is a big concern currently) parasites (e.g. intestinal worms), and prion proteins causing spongiform encephalopathies (such as BSE and Variant Creutzfeldt-Jakob disease (vCJD) in humans). Hence you should refer to pathogenic (rather than microbial) hazards when using overall terminology, and should also mention the potential for these non-microbial hazards, as well.
You note that humans are at risk. This should be briefly expanded to mention the additional risks of immunocompromised humans such as those who are very young, old, with comorbidities or on immunosuppressive (e.g. systemic steroidal) therapy. Risks as greatest for those most exposed but not eliminated for others e.g. who may come into contact during dog walks.
Sustainability
This subheading is missing a number.
You mention the environmental impacts of animal- vs plant-based proteins (refs 129-131). This is good but these references are also very outdated. Much significant new research has further highlighted the sustainability problems created by meat-based pet food (whether conventional, or raw), or significant, and discussed in publications such as those below. Some of should be discussed and cited.
https://journals.plos.org/plosone/article?id=10.1371/journal.pone.0291791
(https://aknight.info/wp-content/uploads/2023/10/Pet-food-sustainability-2023-PNG.png - key results)
https://journals.plos.org/plosone/article?id=10.1371/journal.pone.0291791#sec028 > Consistency with prior studies, esp. those by Su et al, Martens et al, Vale and Vale, Leenstra and Vellinga and Pedrinelli et al. The Alexander et al. study was flawed as described but still indicated significant impacts of meat-based diets.
“Pet foods that incorporate by-products instead of competing directly with human food actually alleviate the environmental impact of the human food system [132].”
This is incorrect (a common misunderstanding) and needs correction. It has been recently shown that the by-product fraction requires more average livestock animals to produce, than the non- by-product fraction. This increases overall numbers of average livestock animals required (by 1.4 for dogs and by 1.9 for cats), and hence, increases environmental impacts. The relevant calculations are included within: https://journals.plos.org/plosone/article?id=10.1371/journal.pone.0291791#sec014 > Average livestock numbers (L) required to supply HC and NHC dietary fractions, for dogs, cats and people.
(This section may help aid understanding: https://journals.plos.org/plosone/article?id=10.1371/journal.pone.0291791#sec028 > Animal by-product use within society.)
General discussion
Given the lack of a ‘Conclusions’ section, this final section serves to include conclusions. Accordingly, a summary should be provided (probably at the start of this section) of the scientific evidence relating to RMBDs of (i) pathogenic hazards to pets and humans, (ii) nutritional deficiencies and imbalances, and (iii) environmental impacts associated with meat-based diets. These are major points, but are barely mentioned in this section, at present. Summaries of these need to be included.
“contamination issues following purchase and storage”
Food preparation should be added – pet guardians can be infected during food preparation.
Other
Dogs are described as carnivores. They are actually omnivores and this should be corrected, throughout.
Comments on the Quality of English LanguageAlthough much English is good, unfortunately numerous small grammatical errors remain. The services of (i) a native English speaker with (ii) a high standard of English (many English speakers lack this) should be sought to review the entire manuscript and highlight improvements warranted.
Abbreviations such as RMBDs should always be defined the 1st time the reader encounters them.
‘veggies’ should be replaced with ‘vegetables’
Author Response
Thank you for your valuable comments and suggestions. We have carefully considered all of them and adapted the manuscript accordingly. All changes in the manuscript are marked using the track changes mode in the adapted version and addressed in this letter. A detailed response to all comments can be found below.
Comment 1: This includes the Methodology. That should be removed to a separate section ‘Methodology’... First, it uses only 1 reliable database (Google Scholar is not reliable). (Readers could also reasonably be concerned about any use of Google in China being more restricted than outside China). This is not adequate. Any language restrictions should be specified. The search terms are not adequate. Normally, a much more comprehensive list of terms is given. E.g. it’s not clear the current search would have detected ‘raw meat feeding of dogs’ and it would have missed Canis familiaris/ Canis lupus familiaris/Felis catus. These deficits are indicated by the list of retrieved studies in Table 4 being significantly shorter.
Response 1: Thank you for your constructive feedback. We have taken your suggestions on board and have now separated the methodology into a new section entitled “Materials and Methods”. Furthermore, we have revised the limitations concerning the database, language restrictions, and search terms, as you recommended. Consequently, 34 additional references have been incorporated into the updated version. The revised section is provided below and can be found on L65-76, p2 of the revised manuscript.
“A comprehensive literature search was conducted using Web of Science (https://ww w.webofscience.com/), ScienceDirect (https://www.sciencedirect.com/), Embase (https: //www.embase.com/), Scopus (https://www.scopus.com/), PubMed (https://pubmed. ncbi.nlm.nih.gov/) and Google Scholar (https://scholar.google.com/) databases. The search was performed using different combinations of the following keywords: RMBD/ raw meat diet, BARF/bones and raw food, dogs/cats/pets, Canis familiaris/Felis catus, with a time frame from January 2000 to December 2024. Only articles with titles and abstracts written in English are included. Studies or cases including hypotheses about RMBDs RMDBs without experiment evidence were excluded, as well as letters, opinions, and mini reviews. An overview of findings is provided below, and classification of research evidence was done using five hierarchy level [25], as explained in Table 1.”
Comment 2: “differing substantially from commercial dry/wet adult maintenance diets” – it would be helpful to give an indicative normal range for these.
Response 2: Thank you for pointing this out. The indicative normal ranges have been added accordingly: “differing substantially from commercial dry (generally 25-35% protein and 10-25% fat) and wet (generally 35-55% protein) adult maintenance diets” (L85-86, p3).
Comment 3: Table 2. before each citation provide author, year – so readers can quickly see, e.g. ‘Smith, 2020 [xx]’. Pls do the same for Tables 3-4. Consider two tables: 1 for dogs, 1 for cats
Response 3: Thank you for your feedback. In response, we have added the authors and years of publication to all references cited in Tables 1 through 6, presented in the format e.g., “[35] Sandri et al., 2016”. Furthermore, we have separated Table 2 into two distinct tables according to your suggestion: Table 2 now specifically covers dogs, and Table 3 specifically covers cats.
Comment 4: [38] – what is inulin?
Response 4: Thank you for pointing this out. We acknowledge that the inclusion of inulin as a fibre supplementation in this study was not directly relevant to the primary focus of our research. Therefore, this element has been removed in the revised version of the manuscript.
Comment 5: “only 2 level-I, 1 level-I” – the latter should be level-II. Table 3. Evidence regarding the benefits of RMBDs. Needs to be renamed to Table 3. Possible benefits of RMBDs.
Response 5: We acknowledge and apologise for the inaccuracies present in the previous submission. We have carefully revised these errors in accordance with your suggestions.
Comment 6: Re: “the definition of the ‘balanced diet’ is easily influenced by the quality of ingredients and social and cultural aspects” This needs to be reworded, because the definition of a nutritionally a complete and balanced diet is one that meets the nutritional requirements of dogs/cats, and those requirements are published by authorities such as FEDIAF, AAFCO, NRC. These biological requirements are independent of social and cultural aspects, etc.
Response 6: We agree with your comment. Therefore, the content has been revised to: “the claim of reduced nutritional imbalance is easily influenced by the quality of ingredients and social/cultural factors ” (L291-292, p8).
Comment 7: The list of studies showing microbial contamination is long, yet significantly incomplete. More than 80 studies have demonstrated pathogenic hazards assoc with raw meat diets (https://sustainablepetfood.info/meat-based-diets/studies-raw-meat/). This should at least be mentioned, even if not all are cited.
Response 7: We are most grateful for your valuable suggestion. We have carefully reviewed the provided link and have selected additional references that are pertinent to the scope of this paper. Consequently, twenty-seven further references concerning pathogenic hazards have been incorporated into the revised manuscript. These additions are detailed in Table 5 of the updated version (p10).
Comment 8: ii. pathogenic hazards are not limited to microorganisms (such as bacteria or protozoa) and can include influenza (avian influenza is a big concern currently) parasites (e.g. intestinal worms), and prion proteins causing spongiform encephalopathies (such as BSE and Variant Creutzfeldt-Jakob disease (vCJD) in humans). Hence you should refer to pathogenic (rather than microbial) hazards when using overall terminology, and should also mention the potential for these non-microbial hazards, as well.
Response 8: We appreciate your feedback. Consequently, the terms “microbial pathogen/contamination” have been revised to “pathogenic hazards” throughout the manuscript, as you suggested.
Further risks associated with pathogenic hazards, including influenza, have been added, along with the inclusion of supporting references: “Campylobacter contamination has been noted in RMBDs as well, particularly based on poultry meat and by-products [78,81,82,104-106]. Raw meat feeding has been also associ-ated with infection of Mycobacterium bovis in dogs and cats, leading to tuberculosis in few cases [107-109]. In addition, when consuming raw meat from poultry or wild birds, dogs and cats are highly possible to contract avian influenza (i.e., H5N1, H5N6), which is cur-rently an ongoing outbreak with several reported cases and multiple deaths [110-113]. Other microbial pathogens in RMBDs included C. perfringens [114,115], C. difficile [114-117], Staphylococcus aureus [83,104], Listeria [80], and Brucellosis [118,119], which are relatively minorly detected and can be reduced through commonly used sterilization methods” (L314-323, p9).
Although less frequently reported in raw pet food, parasites and prion proteins have also been included in the discussion, as you recommended: “Beyond the recognised risk of microbial contamination, other pathogenic hazards associ-ated with the feeding of raw meat to pets include parasites (e.g., intestinal worms) and prion proteins (i.e., implicated in bovine spongiform encephalopathy and variant Creutz-feldt-Jakob disease in humans) [126,143,144]. Despite being less frequently documented and/or studied, these potential risks deserve attention” (L355-360, p10).
Comment 9: You note that humans are at risk. This should be briefly expanded to mention the additional risks of immunocompromised humans such as those who are very young, old, with comorbidities or on immunosuppressive (e.g. systemic steroidal) therapy. Risks as greatest for those most exposed but not eliminated for others e.g. who may come into contact during dog walks.
Response 9: Thank you for your suggestion. We have added the mentioned content accordingly: “Such risks are as greatest for immunocompromised humans, such as those who are very young, old, with comorbidities or on immunosuppressive (e.g., systemic steroidal) therapy [131], but not eliminated for others e.g., who may come into contact during dog walks [131,132]” (L333-336, p9).
Comment 10: This subheading is missing a number.
Response 10: We apologise for the error, which has been adapted.
Comment 11: You mention the environmental impacts of animal- vs plant-based proteins (refs 129-131). This is good but these references are also very outdated. Much significant new research has further highlighted the sustainability problems created by meat-based pet food (whether conventional, or raw), or significant, and discussed in publications such as those below. Some of should be discussed and cited.
Response 11: We are grateful for your valuable suggestions. We have reviewed and incorporated the recommended paper, and a new section summarising relevant data has been added accordingly: “A recent study estimated that globally, dogs and cats consume approximately 7.7% and 1.2% of livestock animals, respectively [164]. A complete transition to nutritionally sound vegan diets worldwide could potentially spare 7.0 billion terrestrial livestock animals from slaughter annually, reduce greenhouse gas emissions by 1.3%, and conserve food energy sufficient to feed 150-190% of the existing pet population or 10% of the global human population [164]” (L423-429, p13).
Comment 12: “Pet foods that incorporate by-products instead of competing directly with human food actually alleviate the environmental impact of the human food system [132].” This is incorrect (a common misunderstanding) and needs correction. It has been recently shown that the by-product fraction requires more average livestock animals to produce, than the non- by-product fraction. This increases overall numbers of average livestock animals required (by 1.4 for dogs and by 1.9 for cats), and hence, increases environmental impacts.
Response 12: Thank you for highlighting this point. We have carefully considered the information you provided, and after reviewing the suggested paper, we agree with your comment. Therefore, the preceding discussion concerning the use of by-products has been removed from the manuscript, due to its controversy and limited research and discussion.
Comment 13: Given the lack of a ‘Conclusions’ section, this final section serves to include conclusions. Accordingly, a summary should be provided (probably at the start of this section) of the scientific evidence relating to RMBDs of (i) pathogenic hazards to pets and humans, (ii) nutritional deficiencies and imbalances, and (iii) environmental impacts associated with meat-based diets. These are major points, but are barely mentioned in this section, at present. Summaries of these need to be included.
Response 13: We appreciate your suggestion and concur with your assessment. Consequently, a new ‘Conclusions’ section, summarizing the aforementioned information, has been added to the manuscript: “In summary, current evidence in dogs and cats indicated that feeding RMBDs may lead to a healthy body weight and condition, an improved stool quality, compositional and functional changes in the gut microbiome, upregulated metabolism of protein and amino acids and/or fat, and may elicit anti-inflammatory and anti-oxidant potentials. However, RMBDs also carry considerable risks, including pathogenic hazards (e.g., bacterial, protozoal, influenza), nutritional imbalances (e.g., high fat content, vitamin and mineral deficiencies), the possibility of oesophageal foreign bodies (i.e., bones), and sustainability issues (e.g., excessive meat consumption, environmental pollution). Crucially, the substantial variations in the types, processing methods, storage, in-gredients used, nutrient content, and animal dietary habits and individual differences may significantly influence the health effects and risks associated with RMBDs. These as-pects are rarely studied in detail and, as such, further investigations are urgently required” (L504-515, p15).
Comment 14: “contamination issues following purchase and storage” Food preparation should be added – pet guardians can be infected during food preparation.
Response 14: Thank you for your suggestion. The relevant content has been added accordingly: “owners are also at risk due to frequent contact with pathogen-carrying pets and contam-inated pet food (i.e., purchase, storage, preparation, and feeding) ” (L332-333, p9).
Comment 15: Dogs are described as carnivores. They are actually omnivores and this should be corrected, throughout.
Response 15: Thank you for pointing this out. We have revised the text to reflect that these statements represent owners’ beliefs/perceptions. We have specified that: “RMBDs are better suited to the purported ‘wild carnivore nature’ of their pets” (L238, p7), and “Owners who favour RMBDs often perceive it as a “natural” feeding method, emphasizing the unprocessed, whole-ingredient nature and the supposed wild carnivore nature of their dogs and/or cats” (L483, p14).
Comment 16: Although much English is good, unfortunately numerous small grammatical errors remain. The services of (i) a native English speaker with (ii) a high standard of English (many English speakers lack this) should be sought to review the entire manuscript and highlight improvements warranted.
Response 16: We acknowledge and apologise for the grammatical errors present in the manuscript. We have now sought the assistance of a native English speaker to thoroughly review and improve the English language throughout the document.
Reviewer 2 Report
Comments and Suggestions for Authors
Well done on your presentation. Your English is mostly excellent. A few insertions of the letter 'a' would occationally be helpful to clarify your meaning. I have highlighted this in my comments.
It is not an easy task to include scientific studies to back up pro and against stances on raw feeding, without introducing bias. Scientific studies are important but cannot ever show everything. You could perhaps mention that a RMBD is the diet that most closely resembles the canine ancestral diet on which canines have evolved over millions of years.
Line 58, and subsequently, should be RMBD not RMDB
Line 84 Another less common form……….This sentence is a bit unclear. Please clarify.
Line 86 This type of diet is intended to provide NOT intends to provide.
Line 115 Nevertheless, the digestibility …….Please clarify this sentence. Do you mean ‘a few studies have also observed’? Insertion of ‘a’ changes the meaning.
Line 118 In addition, the more ideal…….interesting point, but the more ideal body condition is still likely to be associated with the whole experience of eating the raw diet, not just eating less.
Line 127 Insert ‘a’ ……….observed in a few recent gene……….This changes the meaning and I think clarifies what the point you are making.
Line 127/128 Do you mean dry diets or dry diet? If extruded dry diet, delete the ‘an’ at the end of line 127.
Line 152 Please clarify sentence beginning Numerous studies…….in relation to ‘in which investigations……….
Line 156 Please clarify ‘is mostly lacking for felines’ Do you mean is inadequate? scarce?
Line 156 Sentence beginning Investigators have reported……….is a bit clumsy. Better to restructure. In dogs fed raw meat as compared to dogs fed……microbial diversity was reported to be no different, increased or decreased.
Line 159 in those studies better than among.
Line 162 ……….were fed not where fed.
Line 177 may imply potential – all 3 words suggest the same thing! What about - This suggests dysbiosis
Line 203 Remove comma ……..since they are better adapted
Line 212 Sentence a bit clumsy. Please restructure. Do you mean ………differences in nitrogen metabolism in cats fed RMBDs?
Line 261 Insert ‘a’ A few studies reported…………..
Line 261 …………may have a lower risk………….
Line 265 ………remove ‘a’ if metabolic profiles (plural)
Table 3 ………..Insert ‘of’ Lower risk ‘of’ the development of chronic enteropathy
Line 276 Avoid use of ‘acknowledge medical facts’. Replace with ……….’perceived medical facts’, or ‘current medical opinion’.
Line 276 For instance, some proponents claim that………...Teeth fractures from raw bones are very rare, and the rare occurrence does not negate that RMBDs lead to cleaner teeth, as implied by this sentence.
Line 280 The concept of a ‘balanced diet’ and what it involves is subjective and not a fact.
Line 284/5 Although it is challenging………this needs to be conducted………….It is impractical to formulate nutritional content similar to other diet types. This reductionist approach goes against the ethos of raw feeding, and would almost inevitably introduce more processing which would in itself introduce an extra variable.
Line 292 This section implies that the problem is unique to dogs fed RMBDs. You should discuss that processed food is also subject to microbial contamination and that recall of processed diets is not that rare.
Line 321 ……..of such diets (plural)
Line 322 Are non-pathogenic micro-organisms also killed by high-hydrostatic pressure? This might negate positive effects on microbiome and related metabolism.
Line 338-343 Very few cases in these examples. I see later discussed in General discussion line 432-433.
Line 454 They have adapted a high-carbohydrate metabolism. How true is this? They can cope better with carbohydrate digestions, but a high-carb diet is not optimal.
Author Response
Thank you for your valuable comments and suggestions. We have carefully considered all of them and adapted the manuscript accordingly. All changes in the manuscript are marked using the track changes mode in the adapted version and addressed in this letter. A detailed response to all comments can be found below.
Comment 1: You could perhaps mention that a RMBD is the diet that most closely resembles the canine ancestral diet on which canines have evolved over millions of years.
Response 1: Thank you for your suggestion. We have added this content accordingly: “RMBDs are often selected for their perceived natural characteristics and health benefits, such as resembling the canine/feline ancestral diets, being unprocessed...” (L43-44, p1-2), “RMBDs are better suited to the purported “wild carnivore nature” of their pets, as a RMBD is the diet that most closely resembles the canine ancestral diet on which canines have evolved over millions of years [14,15]” (L238-240, p7).
Comment 2: Line 58, and subsequently, should be RMBD not RMDB. Line 115 Nevertheless, the digestibility …….Please clarify this sentence. Do you mean ‘a few studies have also observed’? Insertion of ‘a’ changes the meaning. Line 127 Insert ‘a’ ……….observed in a few recent gene……….This changes the meaning and I think clarifies what the point you are making. Line 127/128 Do you mean dry diets or dry diet? If extruded dry diet, delete the ‘an’ at the end of line 127.
Response 2: We apologise for the error, which has been adapted accordingly.
Comment 3: Line 84 Another less common form……….This sentence is a bit unclear. Please clarify. Line 86 This type of diet is intended to provide NOT intends to provide.
Response 3: Thank you for your comment. The content has been improved: “Another less frequently encountered type of RMBD involves the combination of a premix, comprising grains, fruits, vegetables, vitamins, and minerals, with a raw meat protein source, either as a mixed blend or as a complete, integrated product.t. This type of diet is intended to provide ...” (L92-95, p3).
Comment 4: Line 118 In addition, the more ideal…….interesting point, but the more ideal body condition is still likely to be associated with the whole experience of eating the raw diet, not just eating less.
Response 4: Thank you for highlighting this point. We concur with your assessment and have, accordingly, amended the conclusion from “these studies are therefore limited” to “these need to be further explored in future research” (L132-133, p4).
Comment 5: Line 152 Please clarify sentence beginning Numerous studies…….in relation to ‘in which investigations……….
Response 5: Thank you for pointing this out. The sentence has been improved: “Numerous studies have made great progress in unravelling the exact composition and functionality of the gastrointestinal microbial community in dogs and cats [52,53], and investigations into RMBDs feeding are also beginning to emerge” (L167-168, p6).
Comment 6: Line 156 Please clarify ‘is mostly lacking for felines’ Do you mean is inadequate? scarce?
Response 6: We have revised this sentence accordingly: “data is scarce for felines” (L170, p6).
Comment 7: Sentence beginning Investigators have reported... is a bit clumsy. Better to restructure. In dogs fed raw meat as compared to dogs fed...microbial diversity was reported to be no different, increased or decreased.
Response 7: Thank you for your suggestion. The sentence has been revised accordingly: “In dogs fed raw meat compared to those fed dry or wet diets, microbial diversity has been reported to be not different [37], increased [35,52], or decreased [54,55]” (L170-172, p6).
Comment 8: Line 159 in those studies better than among. Line 162 ……….were fed not where fed. Line 177 may imply potential – all 3 words suggest the same thing! What about - This suggests dysbiosis. Line 203 Remove comma ……..since they are better adapted.
Line 212 Sentence a bit clumsy. Please restructure. Do you mean ………differences in nitrogen metabolism in cats fed RMBDs?
Line 261 Insert ‘a’ A few studies reported…………..
Line 261 …………may have a lower risk………….
Line 265 ………remove ‘a’ if metabolic profiles (plural)
Table 3 ………..Insert ‘of’ Lower risk ‘of’ the development of chronic enteropathy
Line 276 Avoid use of ‘acknowledge medical facts’. Replace with ……….’perceived medical facts’, or ‘current medical opinion’.
Line 321 ……..of such diets (plural)
Response 8: We acknowledge and apologise for the inaccuracies present in the previous submission. We have carefully revised these errors in accordance with your suggestions.
Comment 9: Line 276 For instance, some proponents claim that...Teeth fractures from raw bones are very rare, and the rare occurrence does not negate that RMBDs lead to cleaner teeth, as implied by this sentence.
Response 9: We are most grateful for your valuable suggestion. We agree with your comment, this content was therefore removed from the manuscript.
Comment 10: Line 280 The concept of a ‘balanced diet’ and what it involves is subjective and not a fact.
Response 10: Thank you for your constructive feedback. We have revised the text to reflect that these statements represent public ‘claim’ rather than ‘definition’: “the claim of reduced nutritional imbalance is easily influenced by the quality of ingredi-ents and social/cultural factors” (L292-293, p8).
Comment 11: Line 284/5 Although it is challenging………this needs to be conducted………….It is impractical to formulate nutritional content similar to other diet types. This reductionist approach goes against the ethos of raw feeding, and would almost inevitably introduce more processing which would in itself introduce an extra variable.
Response 11: We appreciate your feedback. We wish to clarify that we are not advocating for the formulation of RMBDs with nutritional content identical to other diet types. Rather, we are emphasising that, for the purposes of rigor and scientific validity, comparisons between different dietary types should be based on comparable ingredients and nutrient content. Without this, it is challenging to discern whether observed results are attributable to differences in ingredients/nutrients or the dietary type itself. Accordingly, we have revised the content to underscore that this consideration is primarily relevant to ‘academic research’: “Crucially, significant differences in the nutritional composition between RMBDs and control diets (e.g., dry, wet) often exist in the studies, blurring the line between the effect of different nutrient content and the effect of the raw diet type itself. While formulating raw diets with similar nutritional content to traditional diets is challenging, this critical comparison must be addressed in future academic research. Such studies should meticulously consider the specific nutrient profiles and ingredients employed, to accurately assess the essential differences between diet types” (L294-301, p7-8).
Comment 12: Line 292 This section implies that the problem is unique to dogs fed RMBDs. You should discuss that processed food is also subject to microbial contamination and that recall of processed diets is not that rare.
Response 12: Thank you for pointing this out. We agree with your comment and have, accordingly, revised the statement by replacing ‘RMBDs’ with ‘pet food’: “pathogenic hazard remains a significant concern in pet food” (L307, p8).
Comment 13: Line 322 Are non-pathogenic micro-organisms also killed by high-hydrostatic pressure? This might negate positive effects on microbiome and related metabolism.
Response 13: We acknowledge your point. Nevertheless, we believe that this does not represent an issue, given that high hydrostatic pressure is only employed during the processing of RMBD products. The microbiome, by contrast, colonises the internal environment of animals. Consequently, its effects, including microbial metabolism, are predominantly exerted within the animal’s gastrointestinal tract, rather than in the food matrix. Additionally, probiotics are generally absent from RMBD formulations and are commonly supplemented independently.
Comment 14: Line 454 They have adapted a high-carbohydrate metabolism. How true is this? They can cope better with carbohydrate digestions, but a high-carb diet is not optimal.
Response 14: Thank you for highlight this point. We agree and have revised the statement to replace ‘high-carbohydrate metabolism’ with ‘a higher carbohydrate metabolism’: “dogs have evolved significantly from their wild ancestors and have adapted to a higher carbohydrate metabolism to meet their nutritional requirements [22]” (L486-487, p14).
Reviewer 3 Report
Comments and Suggestions for Authors
This was an in depth and well written summary of work related to RMBD's fed to pets. The paper was well organized and provided a neutral description of most concerns related to the research of RMBD's.
The primary concern with this manuscript is the placement of digestibility and stool quality data into the section of Public Claims. The research cited by the authors have all demonstrated similar and repeated improvements or similarities in digestibility (compared with extruded/canned pet foods). Most practicing nutritionists and nutrition researchers all agree that RMBD's have sufficient published research to support the claim they are more digestible than processed pet foods. These studies that addressed digestibility of nutrients also demonstrated similar or improved stool quality. Therefore, it is unclear how the authors place these effects in "Public Claims"? It would appear that a more thorough review of stool quality and nutrient digestibility be included in Section 3. This is an area that appears to be minimized by the authors. If the intent is to focus only on studies using the same ingredients, then that should be further explained as to not minimize the data currently available related to nutrient digestibility and RMBD's.
Table 4 should be presented in the same order as written in section 6 with Microbial Pathogens first. The authors should also use the same terminology (Microbial Pathogens or Microbial Contamination) in the text and the table to provide more clarity. Same with Nutritional Imbalance (authors use "risks" in the text, not the table).
Should Sustainability on line 368 be section 6.3?
There have been recent cases of cats consuming raw diets or raw milk, becoming infected with H5N1 bird flu. This should be considered as a possible pathogen, especially for cats as it affects populations globally. This should be included and mentioned at minimum for future research efforts.
Line 200 - spell out "vegetables" and eliminate the use of "veggies"
Author Response
Thank you for your valuable comments and suggestions. We have carefully considered all of them and adapted the manuscript accordingly. All changes in the manuscript are marked using the track changes mode in the adapted version and addressed in this letter. A detailed response to all comments can be found below.
Comment 1: The primary concern with this manuscript is the placement of digestibility and stool quality data into the section of Public Claims. The research cited by the authors have all demonstrated similar and repeated improvements or similarities in digestibility (compared with extruded/canned pet foods). Most practicing nutritionists and nutrition researchers all agree that RMBD's have sufficient published research to support the claim they are more digestible than processed pet foods. These studies that addressed digestibility of nutrients also demonstrated similar or improved stool quality. Therefore, it is unclear how the authors place these effects in "Public Claims"? It would appear that a more thorough review of stool quality and nutrient digestibility be included in Section 3. This is an area that appears to be minimized by the authors. If the intent is to focus only on studies using the same ingredients, then that should be further explained as to not minimize the data currently available related to nutrient digestibility and RMBD's.
Response 1: Thank you for your constructive comments. Our decision not to emphasize this aspect of the results was based on several considerations. Firstly, numerous studies asserting that RMBDs improve digestibility and stool quality are constrained by the use of diets with differing ingredients and nutrient content, which may be the actual drivers of those results, rather than the raw format of the diet itself. Secondly, contradictory findings suggest that processed meat can exhibit higher protein digestibility than RMBDs. Consequently, the notion of improved digestibility and stool quality appears to be more of a ‘public claim’ than supported by robust research.
However, we agree that this necessitates further elaboration, and this has been addressed in the updated version: “in several studies claiming that RMBDs have better digestibility, there are significant differences in the ingredients used and nutrient content in the different diets [38,41,42]; these studies are therefore limited. Moreover, existing evidence have shown inconsistent results regarding digestibility, as a few studies have observed higher protein digestibility in high-temperature and high-pressure processed meat compared to the same type of raw meat (details will be discussed later) [42,43]” (L125-130, p3), “These need to be further explored in future research” (L133-134, p4), and “Crucially, significant differences in the nutritional composition between RMBDs and control diets (e.g., dry, wet) often exist in the studies, blurring the line between the effect of different nutrient content and the effect of the raw diet type itself. While formulating raw diets with similar nutritional content to traditional diets is challenging, this critical comparison must be addressed in future academic research. Such studies should meticulously consider the specific nutrient profiles and ingredients employed, to accurately assess the essential differences between diet types” (L295-302, p8-9).
Comment 2: Table 4 should be presented in the same order as written in section 6 with Microbial Pathogens first. The authors should also use the same terminology (Microbial Pathogens or Microbial Contamination) in the text and the table to provide more clarity. Same with Nutritional Imbalance (authors use "risks" in the text, not the table).
Response 2: We are most grateful for your valuable suggestion. We concur with your comment and have, therefore, adapted the terminology in the tables to ensure consistency with the terminology used throughout the article.
Comment 3: Should Sustainability on line 368 be section 6.3?
Line 200 - spell out "vegetables" and eliminate the use of "veggies"
Response 3: Thank you for pointing this out. We apologise for the errors, which have been adapted accordingly.
Comment 4: There have been recent cases of cats consuming raw diets or raw milk, becoming infected with H5N1 bird flu. This should be considered as a possible pathogen, especially for cats as it affects populations globally. This should be included and mentioned at minimum for future research efforts.
Response 4: We appreciate your suggestion and concur with your assessment. The risk of avian influenza has been added to the updated version, along with 4 references related to RMBDs: “In addition, when consuming raw meat from poultry or wild birds, dogs and cats are highly possible to contract avian influenza (i.e., H5N1, H5N6), which is currently an on-going outbreak with several reported cases and multiple deaths [110-113]” (L319-322, p9).
Cases in cats include: 110. Lee, K.; Lee, E.K.; Lee, H.; Heo, G.B.; Lee, Y.N.; Jung, J.Y.; Bae, Y.C.; So, B.; Lee, Y.J.; Choi, E.J. Highly Pathogenic Avian Influenza A(H5N6) in Domestic Cats, South Korea. Emerg. Infect. Dis. 2018, 24, 2343-2347.
- Marschall, J.; Schulz, B.; Priv-Doz, T.C.H.; Priv-Doz, T.W.V.; Huebner, J.; Huisinga, E.; Hartmann, K. Prevalence of influenza A H5N1 virus in cats from areas with occurrence of highly pathogenic avian influenza in birds. J. Feline Med. Surg. 2008, 10, 355-358.
- Songserm, T., Amonsin, A., Jam-on, R., Sae-Heng, N., Meemak, N., Pariyothorn, N., Pariyothorn, S.; Theamboonlers, A.; Poovorawan, Y. Avian influenza H5N1 in naturally infected domestic cat. Emerg. Infect. Dis. 2006, 12, 681.
- Yu, Z.; Gao, X.; Wang, T.; Li, Y.; Li, Y.; Xu, Y.; Chu, D.; Sun, H.; Wu, C.; Li, S.; et al. Fatal H5N6 avian influenza virus infection in a domestic cat and wild birds in China. Sci. Rep. 2015, 5, 10704.
Round 2
Reviewer 1 Report
Comments and Suggestions for Authors
Thank you for thoroughly addressing the previously points. Minor points remain:
p. 13 revise:
“…conserve food energy sufficient to feed 150-190% of the existing pet population or 10% of the global human population [164].”
i. Should be 160-190% as the pet population refers to the global, not US, population(https://doi.org/10.1371/journal.pone.0291791.t021)
ii. Should be 7% of the global human population (https://doi.org/10.1371/journal.pone.0291791.t020)
Conclusion
Consider revising 1st sentence to:
“In summary, limited current evidence in dogs and cats has suggested that …”
Because the evidence currently appears fairly weak.
Other
There seem to be some errors in numbering of subheadings and occasional minor grammatical errors e.g. a missing space where a new word was inserted. Pls carefully recheck all the text.
Author Response
Thank you so much for your suggestions.
We have revised the mentioned content accordingly: "conserve food energy sufficient to feed 160-190% of the existing pet population or 7% of the global human population" (L425-426, p13), and "In summary, limited current evidence in dogs and cats has suggested that..." (L500, p14).
The subheadings have been renumbered correctly, and we have also thoroughly checked the manuscript for any grammatical or formatting errors.